# Length of hospital stay and associated factors among heart failure patients admitted to the University Hospital in Northwest Ethiopia

**Masho Tigabe Tekle**[ID]*, **Abaynesh Fentahun Bekalu, Yonas Getaye Tefera**[ID]

Department of Clinical Pharmacy, School of Pharmacy, College of Medicine and Health Sciences, University of Gondar, Gondar, Ethiopia

\* mashotigabe@gmail.com

## Abstract

### Background

A prolonged length of hospital stay during heart failure-related hospitalization results in frequent readmission and high mortality. The study was aimed to determine the length of hospital stays and associated factors among heart failure patients.

### Methods

A prospective hospital-based cross-sectional study was carried out to determine the length of hospital stay and associated factors among heart failure patients admitted to the medical ward of the University of Gondar Comprehensive Specialized Hospital from January 2019 to June 2020. Multiple linear regression was used to identify factors associated with length of hospital stay and reported with a 95% Confidence Interval (CI). P-value $\leq$ 0.05 was considered as statistically significant to declare the association.

### Result

A total of 263 heart failure patients (mean age: 51.08 ± 19.24 years) were included. The mean length of hospital stay was 17.29 ± 7.27 days. Number of comorbidities (B = 1.494, p < 0.001), admission respiratory rate (B = -0.242, p = 0.009), serum potassium (B = -1.525, p = 0.005), third heart sound (B = -4.118, p = 0.005), paroxysmal nocturnal dyspnea (B = 2.494, p = 0.004), causes of acute heart failure; hypertensive heart disease (B = -6.349, p = 0.005), and precipitating factors of acute heart failure; infection (B = 2.867, p = 0.037) were significantly associated with length of hospital stay. Number of comorbidities, paroxysmal nocturnal dyspnea, and precipitating factors of AHF specifically infection were associated with a prolonged length of hospital stay.

### Conclusion

Heart failure patients admitted to the medical ward had prolonged hospital stays. Thus, clinicians would be aware of the clinical features contributing to the longer hospital stay and

---

**Data Availability Statement:** Data are available from Figshare at: 10.6084/m9.figshare.20131781.

**Funding:** The author(s) received no specific funding for this work.

**Competing interests:** The authors have declared that no competing interests exist.

**Abbreviations:** ADCHF, Acutely Decompensated Chronic Heart Failure; AF, Atrial Fibrillation; AHF, Acute Heart Failure; CAP, Community-Acquired Pneumonia; CRHD, Chronic Rheumatic Heart Disease; DCMP, Dilated Cardiomyopathy; DVHD, Degenerative Valvular Heart Disease; HF, Heart Failure; HIV, Human Immunodeficiency Virus; HTN, Hypertension; IHD, Ischemic Heart Disease; IQR, Inter Quartile Range; JVP, Jugular Venous Pressure; LOS, Length of Hospital Stay; LVEF, Left Ventricular Ejection Fraction; NYHA, New York Heart Association; PND, Paroxysmal Nocturnal Dyspnea; SBP, Systolic Blood Pressure hospital; SD, Standard Deviation; UOGCSH, University of Gondar Comprehensive Specialized Hospital.

implementation of interventions or strategies that could reduce the heart failure patient's hospital stay is necessary.

## Introduction

Heart Failure (HF) is a complex clinical syndrome that results from any structural or functional impairment of ventricular filling or ejection of blood [1]. It is the leading public health problem associated with frequent hospital admissions, prolonged length of hospital stays (LOS), increased health care costs, and mortality rates [2–4]. Acute Heart Failure (AHF) is the main reason for the hospitalization of HF patients as it is a life-threatening medical condition with a rapid development or change of signs and symptoms which requires urgent diagnosis and treatment [5].

Independent of the presence of co-morbidities and risk factors of cardiovascular diseases, a longer LOS during an initial HF hospitalization has been linked to poor clinical outcomes. This includes exposing patients to suffering from life-threatening medical complications, increased readmission, and high mortality [6–11]. Furthermore, prolonged LOS results in increased use of healthcare resources [9, 12–14].

Several studies described the LOS in HF-associated hospitalization is higher with an estimated median range of 7–21 days [4, 7, 13, 15–21]. The Sub-Saharan Africa Survey of Heart Failure which includes Ethiopian patients showed that the median LOS was 7 days [15]. There was variability in LOS reported by previous studies in Ethiopia with the median duration of hospitalization for HF patients being 11 days at St. Paul's Hospital Millennium Medical College [20] and 4 days at Tikur Anbessa Specialized Hospital [22].

As to the several studies, time spent in the hospital among HF patients is affected by a variety of individual patient's characteristics including socio-demographic variables, clinical presentation at admission, presence of comorbid illness, severity of a disease, in-hospital treatment, and the development of iatrogenic complications. According to these studies being female gender, New York Heart Association (NYHA) functional class, low left ventricular ejection fraction (LVEF), concurrent community-acquired infection, arrhythmias, cerebrovascular disease, dementia, hyponatremia, polypharmacy, pressure ulcers, chronic alcohol consumption, a higher number of comorbidities, peripheral edema, the development of renal impairment, the presence of social problems that need a particular intervention, and concomitant acute medical problems requiring specific treatment were associated with a prolonged LOS [2, 11–14, 21, 23, 24].

The ability to identify hospitalized HF patients at risk of prolonged LOS might be valued by the patients and health care providers. Risk stratification for LOS may help to bring good opportunities for patient care by identifying patients who need special attention and certain interventions such as education and initiation of specific therapies [23]. Nowadays health care professionals are striving to find interventions that help to improve clinical outcomes and reduce healthcare-associated costs. Reduction in the LOS for hospitalized patients has been considered as a primary strategy for efficient resource utilization [25–27].

Despite the growing burden and economic impact of HF in developing countries including Ethiopia, there is either a paucity or inconsistency of data regarding the LOS and associated factors among patients with HF. To the best of literature search, studies are lacking to describe the determinants of LOS for patients hospitalized with HF in Ethiopia. In providing an intervention to reduce the LOS of hospitalized heart failure patients identification and targeting of

the individual heart failure patients' factors which are identified as risk factors for a prolonged LOS is the starting point. In addition to the identification of these factors the finding of this study would provide data for clinicians on which factors they should target in reducing the LOS of their patients. Therefore, this study was conducted to determine the LOS and associated factors among hospitalized AHF patients in the medical ward of the University of Gondar Comprehensive Specialized Hospital (UOGCSH) in Ethiopia.

## Materials and methods

### Study setting and period

The study was conducted in the medical ward of UOGCSH from January 2019 to June 2020. It is located 738 km away from the capital city, Addis Ababa. Currently, it has four major departments namely pediatrics, internal medicine, surgery, gynecology, and obstetrics. Each department has an emergency, inpatient, and outpatient units. The internal medicine inpatient has two main subunits, ward C (serves for female patients) and ward D (serves for male patients). Hospitalized AHF patients are managed in either ward C or ward D. The internal medicine wards are run by nurses, residents, and interns under the supervision of senior physicians.

### Study design

A prospective observational hospital-based cross-sectional study was carried out to determine the LOS and associated factors among hospitalized AHF patients admitted at the UOGCSH medical ward, Northwest Ethiopia.

### Eligibility criteria

**Inclusion criteria.** Patients' $\geq$ 18 years of age.
Diagnosis of AHF.
Admitted to and were discharged alive from the medical ward between January 2019 and June 2020.

**Exclusion criteria.** Patients with missing information regarding the number of days spent in the hospital, Patients with reported LOS < 1 day.

### Sample size determination and sampling procedure

All patients with the diagnosis of AHF attending the medical ward of UOGCSH from January 2019 to June 2020 and those who fulfill the inclusion criteria were taken as the study sample. The study participants were selected by a convenient sampling technique.

### Study variables

**Dependent variable.** Length of hospital stays (days) was described with both mean ($\pm$SD) and median (IQR).

**Independent variables.** The following patient demographic and clinical characteristics were used as independent variables.

Socio-demographic variables: Sex, age.

Causes of AHF: Ischemic heart disease (IHD), hypertension (HTN), atrial fibrillation (AF), stroke, dilated cardiomyopathy (DCMP), degenerative valvular heart disease (DVHD), chronic rheumatic valvular heart disease (CRHD), cor pulmonale, anemia, asthma, chronic obstructive pulmonary disease, thyroid disorder, and renal disease.

Precipitating factors of AHF: DCMP, AF, HTN, IHD, non-adherence to medical regimen, infection (tuberculosis, community-acquired pneumonia), anemia, renal disease, and thyroid disorder.

Co-morbidity: Human immune deficiency virus (HIV), IHD, AF, CRHD, DVHD, community-acquired pneumonia (CAP), chronic obstructive pulmonary disease, dyslipidemia, renal disease, cor pulmonale, diabetes mellitus, liver disease, stroke, thyroid disease, asthma, cancer, tuberculosis, anemia.

The number of comorbidities.

Clinical presentation at admission: NYHA class, paroxysmal nocturnal dyspnea (PND), AF, systolic blood pressure (SBP), diastolic blood pressure, respiratory rate, heart rate, temperature, LVEF, serum creatinine, hemoglobin, serum sodium, serum potassium, dyspnea at rest, dyspnea on exertion, easy fatigability, peripheral edema, elevated jugular venous pressure (JVP), cardiomegaly, third heart sound, orthopnea, neck vein distension, murmur.

Cardiac medications: used for management of AHF during the hospital stay and at discharge.

The number of medications during the hospital course.

Mechanical ventilation: use of oxygen therapy during the hospital stay.

## Data collection procedures

By using a pretested abstraction format (S1 Annex) data were collected by three trained nurses. Data abstraction format was prepared by reviewing similar studies [2, 8, 21, 23, 24, 28]. Data regarding demographic variables, medical history, clinical presentation on admission, echocardiographic and laboratory findings, and in-hospital treatment were collected through medical chart review and recorded on the data abstraction format.

## Data quality control technique

To assure the completeness of the data abstraction format, a pre-test was conducted in the emergency medical in-patients and proper modification was employed to the format.

## Data entry and statistical analysis

Data was edited; cleaned, coded, entered, and analyzed using Statistical Package for Social Sciences version 21. Descriptive, correlation, comparative, and regression analysis were conducted. Continuous variables were expressed as mean (±SD) when normally distributed or median (IQR) when not normally distributed. Additionally, categorical variables were summarized as frequency (percentage) of the total. Since the dependent variable LOS was a continuous variable which fulfill the normality distribution and linearity assumptions; multiple linear regression was used to identify factors associated with LOS. P-value $\leq 0.05$ was considered as statistically significant to declare the association. In univariate analyses, Pearson and Spearman's rank correlation coefficients were used to determine the correlation between variables. Multicollinearity was assessed using Pearson correlation coefficients. Comparative analysis was conducted on the LOS across different socio-demographic, clinical, and in-hospital treatment-related characteristics of patients using independent sample t-test and One-way Analysis of Variance (ANOVA) test. A skewness test was used for checking normality. In the study most of the independent variables were categorical variables which were categorized as Yes/No while few variables such as age, number of comorbidities, number of medications, were continuous. Additionally, during analysis categorical variables with more than two categories specifically precipitating factors of AHF and causes of AHF were analyzed by converting them in to dummy variables (Yes/No).

## Operational definition

HF is a clinical syndrome characterized by typical symptoms (e.g. breathlessness, ankle swelling, and fatigue) that may be accompanied by signs (e.g. elevated jugular venous pressure, pulmonary crackles, and peripheral edema) caused by a structural and/or functional cardiac abnormality, resulting in reduced cardiac output and/ or elevated intracardiac pressures at rest or during stress [5].

AHF refers to the rapid onset or worsening of symptoms and/or signs of HF [5].

New-onset HF: AHF occurs in patients without a history of HF [5].

Acutely decompensated chronic heart failure (ADCHF): AHF occurs in patients with a history of chronic HF [5].

Length of hospital stays: was defined as the difference between the discharge and admission dates.

## Ethics approval and consent to participate

Ethical clearance was obtained from the University of Gondar, College of Medicine and Health Sciences, School of Pharmacy Ethical Clearance Committee. Data was collected mainly through medical chart review and informed oral consent was obtained from each patient involved in the study. In addition, the study was done as per the declaration of Helsinki.

## Result

### Socio-demographic and clinical characteristics of patients

Between January 2019 and June 2020, a total of 290 HF patients were admitted to the medical ward of UOGCSH. Of these 263 HF patients who fulfill the inclusion criteria were included in this study. The mean (±SD) age was 51.08 (±19.24) years, 153 (58.2%) were females and 58.9% of patients had a new-onset type of HF. In the study, 88 (33.5%) of the patients were admitted with an undefined precipitating factor. Of the defined ones, CAP 79 (30%) and AF 52 (19.8%) were the leading precipitating factors for AHF. Among the ADCHF patients', non-adherence to the medical regimen 33 (30.5%) was the leading precipitating factor followed by CAP 25 (23.1%). Fifty-one percent of patients had a prior history of CAP, 31.2% had IHD, 29.3% had AF, 27% had HTN, and 26.6% had CRHD. The mean (±SD) number of comorbidities was 3.34 (±1.42). The mean (±SD) LVEF was 46 (±20.55) (Table 1).

On admission, 236 (89.7%), 225 (85.6%), 210 (79.8%), and 190 (72.2%) of patients were presented with dyspnea on exertion, dyspnea at rest, peripheral edema, and orthopnea, respectively (Fig 1). The result showed that IHD was the most common underlying cause of AHF (28.5%) which was followed by CRHD (19.4%) (Fig 2).

### In-hospital treatment of AHF and discharge medications

The median (IQR) number of medications taken during the hospital course was 3 (1–4). During hospitalization, almost all patients were treated with furosemide 260 (98.9%). Next to furosemide the most frequently prescribed medications were aspirin 98 (37.3%), atorvastatin 97 (36.95), and spironolactone 73 (27.8%). At discharge, furosemide, spironolactone, and enalapril were prescribed for 198 (75.3%), 80 (30.4%), and 63 (24.0%) patients, respectively (Fig 3). Additionally, mechanical ventilation via intranasal oxygen therapy was used in 90 (34.2%) of patients.

### Length of hospital stay

The LOS data were normally distributed. The mean (±SD) and median (IQR) LOS were 17.29 (±7.27) days and 18 (12–23) days, respectively. There were 138 (52.5%) patients with LOS ≥ 18 days and 125 (47.5%) patients with LOS ≤ 17 days.

**Table 1. Socio-demographics and clinical characteristics of HF patients admitted at UOGCSH medical ward.**

| Variable | | Frequency (%) | Mean (±SD) | Median (IQR) |
|---|---|---|---|---|
| Age (years) | | | 51.08 (± 19.24) | |
| Gender | | | | |
| | Male | 110 (41.8) | | |
| | Female | 153 (58.2) | | |
| Vital signs | | | | |
| | SBP (mmHg) | | | 110 (100–130) |
| | Diastolic blood pressure (mmHg) | | | 70 (60–80) |
| | Heart rate (beats/minute) | | | 96 (88–110) |
| | Respiratory rate (breaths/minute) | | | 24 (22–28) |
| | Temperature (C$^0$) | | | 36.60 ± 0.04 |
| Laboratory findings | | | | |
| | Serum creatinine (mg/dl) | | | 0.89 (0.71-.89) |
| | Serum sodium (meq/L) | | 139.1 (± 5.68) | |
| | Serum potassium (meq/L) | | | 3.84 (3.4–4.36) |
| | Hemoglobin (g/dl) | | | 12.6 (0.71–1.3) |
| Comorbidities | | | | |
| | CAP | 134 (51) | | |
| | IHD | 82 (31.2) | | |
| | AF | 77 (29.3) | | |
| | HTN | 71 (27) | | |
| | Anemia | 62 (23.6) | | |
| | DVHD | 70 (26.6) | | |
| | Cor pulmonale | 59 (22.4) | | |
| | CRHD | 48 (18.3) | | |

IHD, Ischemic Heart Disease; DVHD, Degenerative Valvular Heart Disease; CRHD, Chronic Rheumatic Heart Disease; CAP, Community-Acquired Pneumonia; HTN, Hypertension; AF, Atrial Fibrillation; SBP, Systolic Blood Pressure.

Length of hospital stay showed a moderate positive association with the number of comorbidities (p < 0.001, Pearson's r = 0.286). On the other hand, the number of prescribed medications during the hospital course (p = 0.004, Spearman's r = -0.175) and presence of elevated JVP (p = 0.02, Spearman's r = -0.143) at admission had a weak negative association with LOS (S1 Table).

Patients hospitalized due to CRHD as a cause of AHF-related admission showed a higher mean (± SD) of 19.75 (±8.28) days of LOS than other causes (p = 0.005). Compared to other precipitating factors of HF, CAP showed higher mean ± (SD) 20.30 (±8.02) days (p < 0.001) of LOS (Table 2).

## Predictors of length of hospital stay

The multiple linear regression indicated that number of comorbidities (B = 1.494, p < 0.001), admission respiratory rate (B = -0.242, p = 0.009), serum potassium (B = -1.525, p = 0.005), third heart sound (B = -4.118, p = 0.005), PND (B = 2.494, p = 0.004), causes of AHF specifically hypertensive heart disease (B = -6.349, p = 0.005), and precipitating factors of AHF specifically infection (B = 2.867, p = 0.037) were significantly associated with LOS.

For every unit increase in the number of comorbidities, there is 1.494 increase in the LOS. Patients who were admitted with the presence of PND and the precipitating factors of AHF specifically infection had 2.494 and 2.867 times higher LOS, respectively, comparing to their

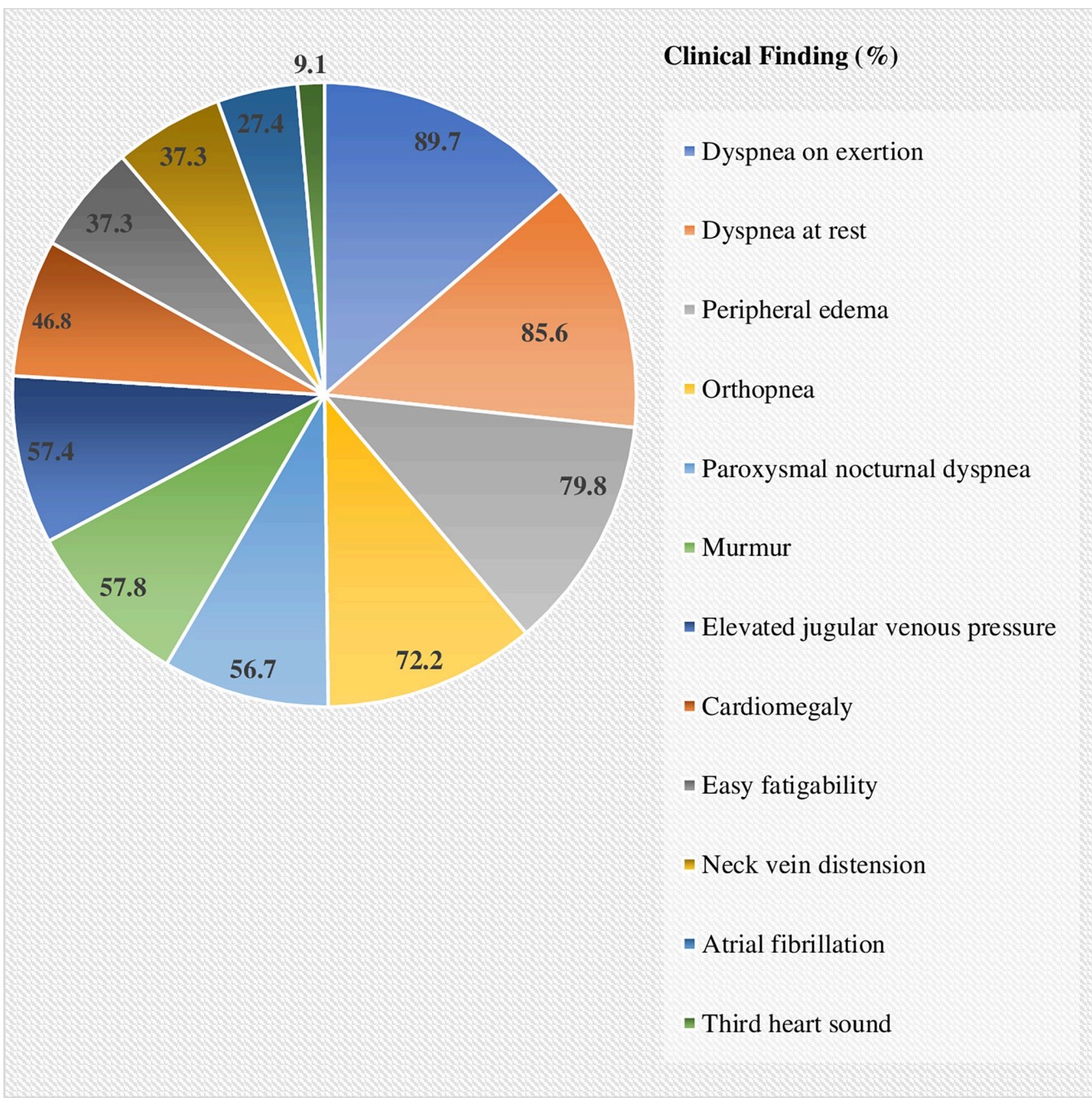

**Fig 1. Clinical findings of HF patients admitted at UOGCSH medical ward, 2020 (N = 263).**

counterparts. In contrast to this, the study showed that for every unit increase in the admission respiratory rate and serum potassium there is 0.242 and 1.525 decrease in the LOS, respectively. Comparing to patients who were admitted without third heart sound, patients presented with third heart sound had 4.118 times lower LOS. Additionally, comparing to other causes of AHF patients who were admitted because of hypertensive heart disease had 6.349 times lower LOS.

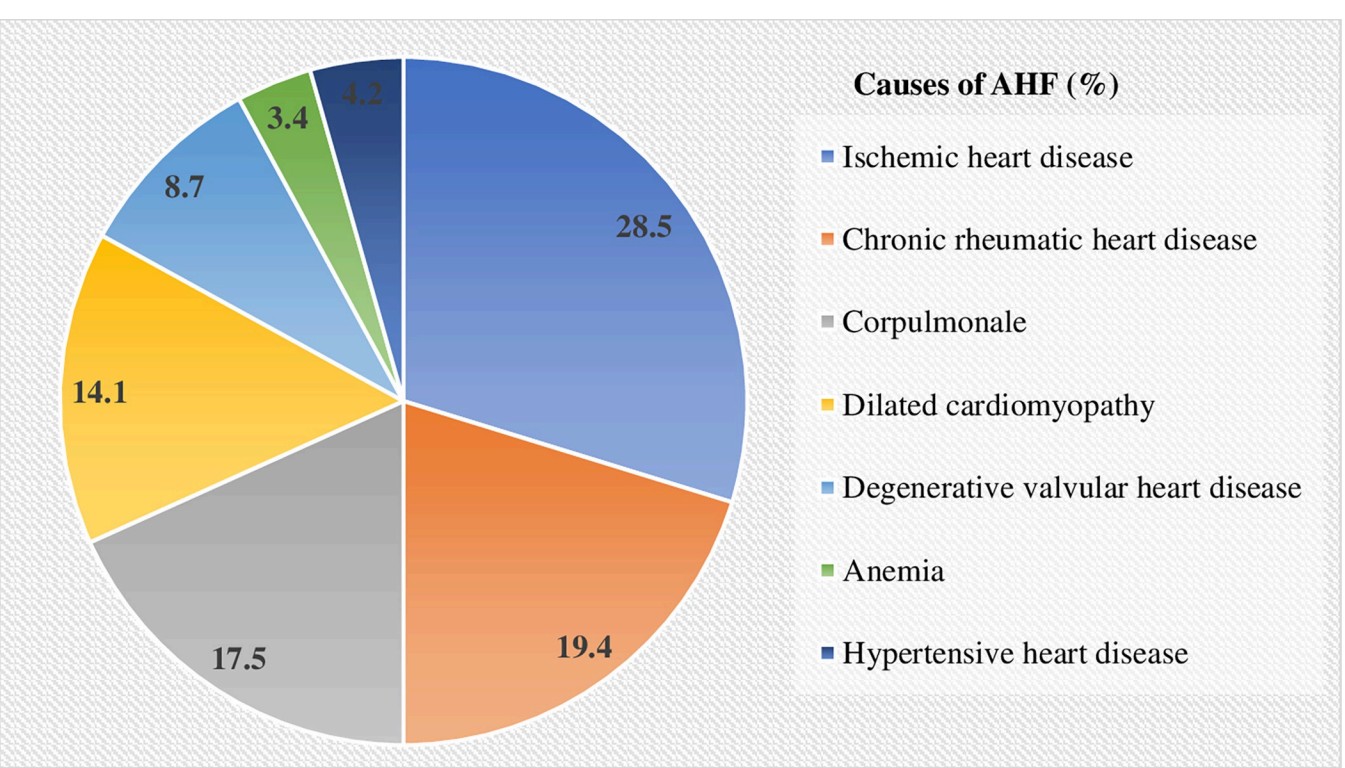

**Fig 2. Causes of AHF among HF patients admitted at UOGCSH medical ward, 2020 (N = 263).**

The model analysis showed that the independent variables explain 35.9% of the variability of dependent variable LOS ($R^2$ = 0.359, adjusted R square = 0.286) and the regression model was a good fit of the data, (F = 27,235) = 4.881, p < 0.001 (Table 3).

## Discussion

The results of the present study showed that the mean (±SD) LOS was 17.29 (±7.27) days. According to the multiple linear regression increased number of comorbidities, PND, and precipitating factors of AHF specifically infection were significantly associated with a prolonged length of hospital stay.

### Socio-demographic and clinical characteristics of patients

In the present study, the mean age of the participants was 51 years which was relatively lower compared to previous studies (71–73 years) [2, 8, 24]. The discrepancy might be explained by these studies were conducted in developed countries where the mean age of their adult population is relatively higher than the developing countries, but was comparable with the Sub-Saharan Africa Survey of Heart Failure [15] and other HF studies (47–53 years) conducted in Ethiopia [3, 20]. This study had shown that 58.2% of patients were females and similar to an observational study conducted at Tikur Anbessa Specialized Hospital (54.4%) [22].

According to the findings of this study, all patients were presented with either in NYHA class III (1.9%) or IV (98.1%). This is different from the proportion reported by a retrospective observational cohort study in Japan (NYHA class III = 35.4%, IV = 28.5%) [21] and a prospective study from Hospital Universitari de Bellvitge (NYHA Class III = 17%, IV = 76.9%) [12]. These differences could be explained as in the present study majority of patients were

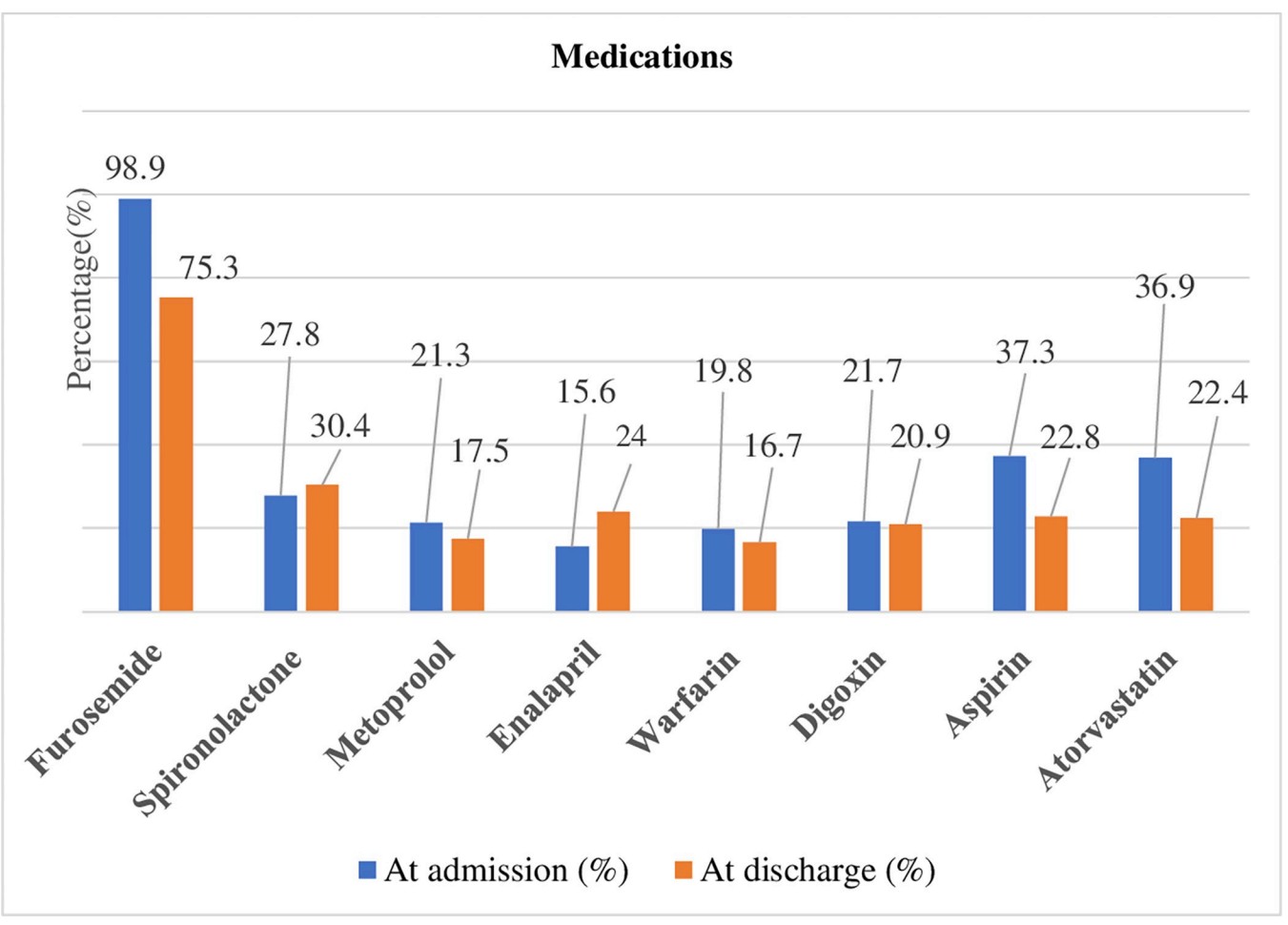

**Fig 3. Medications received by AHF patients admitted at UOGCSH medical ward, 2020 (N = 263).**

presented with more severe form of HF as documented with the presence of dyspnea on exertion (89.7%), dyspnea at rest (89.7%), peripheral edema (79.8%), and orthopnea (72.2%).

The present study showed that a third of HF patients were admitted with an undefined precipitating factor. Of the defined ones, CAP (30%) and AF (19.8%) were the leading precipitating factors for AHF. Similarly, a study from St Paul's Hospital Millennium Medical College in Ethiopia reported that 25.6% of patients were hospitalized without a definitive cause of precipitating factors of HF [20]. Additionally, CAP (28%), infective endocarditis (10.9%), and AF (8.7%) were the most frequent precipitating factors for AHF in this study [20]. Among the ADCHF, non-adherence to the medical regimen and CAP were the leading precipitating factors in nearly a third (30.5%) and a quarter (23.1%) of cases, respectively. Similarly, the Ugandan study identified that non-adherence to medical regimens and infection were the precipitating factors for HF decompensation in 31.7% and 26.7% of patients, respectively [29].

## In-hospital treatment of AHF and discharge medications

Furosemide was administered to almost all (98.9%) HF patients during hospitalization. This was consistent with a study done at Tikur Anbessa Specialized Hospital in Ethiopia which revealed that furosemide was prescribed to 95.9% of hospitalized HF patients [22]. Similarly, results from the Italian and Sub-Saharan Africa survey of heart failure showed that furosemide

**Table 2. Association of clinical factors with LOS among AHF patients admitted at UOGCSH medical ward.**

| Characteristics | | | Length of hospital stay | |
| --- | --- | --- | --- | --- |
| | | | Mean (±SD) | P-value |
| Types of HF | New-onset | | 17.58 (±7.18) | 0.437[a] |
| | ADCHF | | 16.87 (±7.41) | |
| Causes of HF | | | | 0.005[b] |
| | IHD | | 15.80 (±7.16) | |
| | Hypertensive heart disease | | 11.18 (±6.4) | |
| | DCMP | | 18.57 (±6.72) | |
| | DVHD | | 16.65 (±7.72) | |
| | CRHD | | 19.75 (±8.27) | |
| | Cor pulmonale | | 18.09 (±6.280) | |
| | Anemia | | 18.11 (±5.13) | |
| Comorbidities | | | | |
| | DVHD | | | 0.046[a] |
| | | Yes | 18.77 (±8.45) | |
| | | No | 16.75 (±6.74) | |
| | HIV | | | 0.003[a] |
| | | Yes | 24.33 (±8.82) | |
| | | No | 17.04 (±7.11) | |
| | CRHD | | | 0.006[a] |
| | | Yes | 19.88 (±8.66) | |
| | | No | 16.71 (±6.82) | |
| | CAP | | | 0.001[a] |
| | | Yes | 18.7 (±7.46) | |
| | | No | 15.82 (±6.79) | |
| Number of comorbidities | | | | < 0.001[a] |
| | | ≤ 3 | 15.48 (±6.83) | |
| | | ≥ 4 | 19.65 (±7.19) | |
| Clinical findings | | | | |
| | Elevated JVP | | | 0.017[a] |
| | | Yes | 16.37 (±6.88) | |
| | | No | 18.53 (±7.63) | |
| | PND | | | 0.007[a] |
| | | Yes | 16.23 (6.54) | |
| | | No | 18.67 (7.94) | |
| | Third heart sound | | | 0.001[a] |
| | | Yes | 21.79 (±7.44) | |
| | | No | 16.84 (±7.12) | |
| | Murmur | | | 0.016[a] |
| | | Yes | 18.21 (±7.53) | |
| | | No | 16.03 (±6.73) | |
| | SBP (mmHg) | | | 0.003[b] |
| | | ≤119 | 17.77 (±7.20) | |
| | | 120–139 | 17.80 (±6.85) | |
| | | 140–159 | 16.86 (±7.58) | |
| | | ≥ 160 | 10.43 (±5.85) | |
| Laboratory values | | | | |
| | Serum potassium | | | < 0.001[b] |

(*Continued*)

**Table 2.** (Continued)

| Characteristics | | | Length of hospital stay | |
|---|---|---|---|---|
| | | ≤ 3.549 | 19.94 (±7.14) | |
| | | 3.55–5.55 | 16.42 (±6.95) | |
| | | > 5.55 | 12.27 (±7.84) | |
| Medication | | | | |
| | Enalapril | | | 0.076[a] |
| | | Yes | 15.44 (±6.51) | |
| | | No | 17.63 (±7.37) | |
| | Atenolol | | | 0.019[a] |
| | | Yes | 20.86 (±7.96) | |
| | | No | 16.98 (±7.14) | |
| | Dopamine | | | 0.022[a] |
| | | Yes | 21.31 (±8.88) | |
| | | No | 17.03 (±7.09) | |
| | Atorvastatin | | | 0.007[a] |
| | | Yes | 15.71 (±6.89) | |
| | | No | 18.21 (±7.35) | |
| Number of medications | | | | 0.012[a] |
| | | ≤ 3 | 18.23 (±7.13) | |
| | | ≥ 4 | 15.95 (±7.29) | |

ADCHF, Acutely Decompensated Chronic Heart Failure; IHD, Ischemic Heart Disease; DCMP, Dilated Cardiomyopathy; DVHD, Degenerative Valvular Heart Disease; CRHD, Chronic Rheumatic Heart Disease; HIV, Human Immunodeficiency Virus; CAP, Community-Acquired Pneumonia; JVP, Jugular Venous Pressure; PND, HTN, Hypertension; AF, Atrial Fibrillation; Paroxysmal Nocturnal Dyspnea; SD, Standard Deviation; IQR, Inter Quartile Range.

a an Independent t-test

b Analysis of variance (ANOVA).

was used by 98.1% and 92.9% of patients, respectively [15, 17]. At hospitalization, aspirin was administered to 36.95% of patients. This is comparable to the Ugandan study in which aspirin was prescribed in 32.9% of patients [29]. Like that of aspirin, the rate of atorvastatin use during hospitalization was high (27.8%). In the HF-Turkey study, the rate of use of statin at admission was 24.7% which was comparable with the present study [30]. Generally, the frequent prescription of aspirin and atorvastatin in the current study might be due to the existence of IHD as a leading co-morbid condition since these medications are recommended as its treatment of choice [31, 32]. At discharge, furosemide (75.3%), spironolactone (30.4%), and enalapril (24.0%) were the most widely used medications. These findings are similar to previous studies which described furosemide, spironolactone, and enalapril as the most common discharge medications for HF patients [20, 21].

## Length of hospital stay

In the current study, the mean (±SD) LOS of the hospitalized HF patients was 17.29 (±7.27) days. This was in agreement with a retrospective cohort study carried out in Japan 19.5 (±12.5 days) [21], however, it was higher than a retrospective cohort study conducted in California 3.8 (± 4.8) days [8], Gulf Acute Heart Failure Registry 3 (±12.5) days [13], and Italian study 11.2 (± 6.7) days [17]. The median (IQR) length of hospitalization was 18 (12–23) days and comparable to the Japanese study which was 17 (11–25) days [21] and West Tokyo Heart Failure registry was 15 (10–23) days [4]. However, it was higher than a cohort study conducted in

**Table 3. Multiple linear regression for LOS of AHF patients admitted at UOGCSH medical ward.**

| Variables | B | Standard error | ß | T | P-value | 95%CI (upper, lower) |
|---|---|---|---|---|---|---|
| Age | -0.010 | 0.029 | -0.27 | -0.354 | 0.723 | (-0.068,0.047) |
| Hypertensive heart disease -causes of AHF | -6.349 | 2.265 | -0.175 | -2.803 | 0.005* | (-10.811,1.886) |
| Dilated cardiomyopathy—causes of AHF | -0.016 | 1.483 | -0.001 | -0.011 | 0.991 | (-2.938,2.906) |
| DVHD—causes of AHF | -2.505 | 2.066 | -0.098 | -1.212 | 0.227 | (-6.577,1.566) |
| CRHD- causes of AHF | 0.025 | 1.635 | 0.001 | 0.015 | 0.988 | (-3.197,3.247) |
| Corpulmonale—causes of AHF | -0.414 | 1.453 | -0.022 | -0.285 | 0.776 | (-3.277,2.448) |
| Anemia—causes of AHF | -0.827 | 2.475 | -0.021 | -0.334 | 0.378 | (-5.703,4.048) |
| Others—causes of AHF | -0.597 | 2.256 | -0.016 | -0.264 | 0.792 | (-5.040,3.847) |
| Atrial fibrillation–precipitating factor of AHF | -0.660 | 1.1175 | -0.036 | -0.562 | 0.575 | (-2.975,1.655) |
| Non-compliance to medical regimen—precipitating factor of AHF | 0.436 | 1.325 | 0.022 | 0.329 | 0.743 | (-2.176,3.047) |
| Infection—precipitating factor of AHF | 2.867 | 1.367 | 0.181 | 2.098 | 0.037* | (0.175,5.560) |
| DVHD | 1.014 | 1.422 | 0.062 | 0.713 | 0.477 | (-1.789,3.816) |
| HIV | -4.272 | 2.270 | -0.107 | -1.882 | 0.061 | (-8.744,0.200) |
| CAP | 0.857 | 1.197 | 0.059 | 0.716 | 0.475 | (-1.501,3.215) |
| Number of comorbidities | 1.494 | 0.357 | 0.291 | 4.186 | 0.000** | (0.791,2.196) |
| NYHA class | 3.090 | 3.101 | 0.058 | 0.996 | 0.320 | (-3.020,9.201) |
| SBP | 0.062 | 0.034 | 0.191 | 1.792 | 0.074 | (-0.006,0.129) |
| Diastolic blood pressure | -0.082 | 0.043 | -0.181 | -1.912 | 0.057 | (-0.167,0.002) |
| Respiratory rate | -0.242 | 0.091 | -0.157 | -2.650 | 0.009* | (-.422,-0.062) |
| Serum potassium | -1.525 | 0.543 | -0.171 | -2.807 | 0.005* | (-2.596,-0.455) |
| Elevated JVP | 1.604 | 0.850 | 0.109 | 1.887 | 0.060 | (-0.070,3.279) |
| Third heart sound (S$_3$gallop) | -4.118 | 1.466 | -0.163 | -2.809 | 0.005* | (-7.006,-1.230) |
| PND | 2.494 | 0.857 | 0.170 | 2.911 | 0.004* | (0.806,4.181) |
| Murmur | -1.017 | 0.918 | -0.069 | -1.108 | 0.269 | (-2.825,0.792) |
| Atenolol | -0.871 | 1.543 | -0.033 | -0.564 | 0.573 | (-3.912,2.170) |
| Dopamine | -2.100 | 1.846 | -0.069 | -1.138 | 0.256 | (-5.736,1.537) |
| Number of medications | -0.494 | 0.253 | -0.132 | -1.952 | 0.052 | (-0.993,0.005) |

AHF, Acute Heart failure, NYHA, New York Heart Association; SBP, Systolic Blood Pressure; DVHD, Degenerative Valvular Heart Disease; HIV, Human Immunodeficiency Virus; CAP, Community-Acquired Pneumonia; JVP, Jugular Venous Pressure; PND, Paroxysmal Nocturnal Dyspnea; B, Unstandardized Coefficients; ß, Standardized Coefficients; t, t-statistic.

*p-value ≤ 0.05

**p-value < 0.001.

New Zealand; 6 (4–9) days [2], Italian study; 10 (7–14) days [17], Sub-Saharan Africa Survey of Heart Failure registry; 7 (5–10) days [15] and the American Get With The Guidelines-HF registry; 4 (2–6) days [23].

## Factors associated with length of hospital stay

In the current study, patients with prolonged LOS were more likely to be presented with a higher number of comorbidities. Similarly, the Get with the Guidelines-HF registry revealed that patients with LOS > 7 days have presented to the hospital with a greater number of comorbidities compared to those with < 7 days of LOS [23]. A cross-sectional study in the US also identified a higher number of comorbidities were associated with a longer LOS in AHF patients [14]. Additionally, this study showed that patients with prolonged LOS were more likely to be presented with a sign and symptoms of congestion specifically PND. A study done in New Zealand also showed that peripheral congestion was independently associated with

longer (> 10 days) LOS [2]. Despite the simple linear regression demonstrating that CAP was associated with LOS, the multiple regression showed that CAP was not significantly associated with LOS. In contrast to this finding, a retrospective cohort study in Japan reported that pneumonia was strongly associated with prolonged LOS [21]. This might be attributed to a large number of AHF patients were included in the Japanese study.

To the best of the authors' literature search, this prospective study was the first to describe LOS and associated factors among hospitalized HF patients in Ethiopia. Despite its' strength, it is not without limitations. Firstly, it involves a single population who were stratified by a specific disease criterion which was HF. However, the LOS among patients with the same disease may vary owing to complex factors related to the individual patient or organizational factors or divergences in the medical practice. Secondly, compared to the other studies the sample size in the present study was relatively small and significant associations could be missed out. Thirdly, this study was conducted in a single health facility. Fourthly, LOS might be influenced by the different health insurance systems, hospital capacity, patterns of clinical practice, and physicians however because of financial and time constraints we can't include these factors. Thus, the generalizability of the results might be limited.

## Conclusion

In conclusion, the present study revealed that HF patients admitted to the medical ward spent prolonged LOS. Patients who were presented with concurrent DVHD, increased number of comorbidities, presence of elevated JVP, and PND were associated with longer LOS. Thus, clinicians would be aware of the clinical features contributing to the longer hospital stays. Implementation of interventions or strategies that could reduce the time spent in hospital is necessary for HF patients. Additionally, these data may help health care providers in identifying patients who are more likely to spent prolonged LOS. Moreover, a study with larger sample size and multicenter approach will be required to further strengthen the current findings.

## Supporting information

**S1 Table. Correlations between patient characteristics and LOS among AHF patients.**
(PDF)

**S1 Annex. Data abstraction format of AHF patients.**
(PDF)

## Acknowledgments

The authors acknowledge the School of Pharmacy, University of Gondar, all the staff of the Departments of the medical ward for their cooperation during the conduct of the study, and the study participants for their willingness to be involved in the research project.

## Author Contributions

**Conceptualization:** Masho Tigabe Tekle, Abaynesh Fentahun Bekalu, Yonas Getaye Tefera.

**Data curation:** Masho Tigabe Tekle, Abaynesh Fentahun Bekalu, Yonas Getaye Tefera.

**Formal analysis:** Masho Tigabe Tekle.

**Methodology:** Masho Tigabe Tekle.

**Writing – original draft:** Masho Tigabe Tekle, Abaynesh Fentahun Bekalu, Yonas Getaye Tefera.

**Writing – review & editing:** Masho Tigabe Tekle, Abaynesh Fentahun Bekalu, Yonas Getaye Tefera.

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
