## [Editor Report · Decision Letter 0]

29 Jan 2021

PONE-D-21-01805

Length of hospital stay and associated factors among heart failure patients admitted to the medical ward of University of Gondar Comprehensive Specialized Hospital, Northwest Ethiopia: A Cross-sectional study

PLOS ONE

Dear Dr. Tigabe,

Thank you for submitting your manuscript to PLOS ONE. After careful consideration, we feel that it has merit but does not fully meet PLOS ONE’s publication criteria as it currently stands. Therefore, we invite you to submit a revised version of the manuscript that addresses the points raised during the review process.

Please do English editing, title revision, and then re-submit. 

We look forward to receiving your revised manuscript.

Kind regards,

Academic Editor

PLOS ONE

Additional Editor Comments:

1. Please revise and shorten the Title to make it concise and informative.

2. Please do English editing first--too many English errors.

2. Please upload a copy of Supporting Information Table S1 and Annex S1 which you refer to in your text on page 28.
---

## [Author Response · Author response to Decision Letter 0]

24 Feb 2021

Rebuttal letter

Dear editor,

Thank you for your sincere comments. Please find our response to your concerns in the previously submitted manuscript in PLOS ONE Journal. 

1. Please revise and shorten the Title to make it concise and informative.

Dear editor, the title was revised and shortened from “Length of hospital stay and associated factors among heart failure patients admitted to the medical ward of University of Gondar Comprehensive specialized Hospital, Northwest Ethiopia: A Cross sectional study” to “Length of hospital stay and associated factors among heart failure patients admitted to the University Hospital in Northwest Ethiopia” 

2. Please do English editing first--too many English errors.

Dear editor, We have made English editing to the best of our capacity and please consider our revised manuscript to proceed in your esteemed journal.

---

## [Decision Letter · Decision Letter 1]

16 Mar 2021

PONE-D-21-01805R1

Length of hospital stay and associated factors among heart failure patients admitted to the University Hospital in Northwest Ethiopia

PLOS ONE

Dear Dr. Tigabe,

Thank you for submitting your manuscript to PLOS ONE. After careful consideration, we feel that it has merit but does not fully meet PLOS ONE’s publication criteria as it currently stands. Therefore, we invite you to submit a revised version of the manuscript that addresses the points raised during the review process.

Please revise accordingly. 

We look forward to receiving your revised manuscript.

Kind regards,

Academic Editor

PLOS ONE

Journal Requirements:

Reviewers' comments:

Reviewer's Responses to Questions

**Comments to the Author**

1. If the authors have adequately addressed your comments raised in a previous round of review and you feel that this manuscript is now acceptable for publication, you may indicate that here to bypass the “Comments to the Author” section, enter your conflict of interest statement in the “Confidential to Editor” section, and submit your "Accept" recommendation.

Reviewer #1: All comments have been addressed

Reviewer #2: All comments have been addressed

2. Is the manuscript technically sound, and do the data support the conclusions?

Reviewer #1: Yes

Reviewer #2: Yes

3. Has the statistical analysis been performed appropriately and rigorously? 

Reviewer #1: Yes

Reviewer #2: Yes

4. Have the authors made all data underlying the findings in their manuscript fully available?

Reviewer #1: Yes

Reviewer #2: Yes

5. Is the manuscript presented in an intelligible fashion and written in standard English?

Reviewer #1: Yes

Reviewer #2: Yes

6. Review Comments to the Author

Reviewer #1: The Authors have adequately replied to previous editor comments. The paper is interesting, however it's not easy to read this manuscript, the intro is too long and also data presentation is complex with a lot of data and numbers. It's unclear why the use of diuretics, the type of underlying disease (i.e CAD, VHD or DCM), the time from symptom onset to hospital admission and finally the age, didn't enter in the multivariate analysis

Reviewer #2: (No Response)

7. PLOS authors have the option to publish the peer review history of their article (what does this mean?). If published, this will include your full peer review and any attached files.

Reviewer #1: **Yes: **luciano agati

Reviewer #2: **Yes: **Shady Abohashem, MD

---

## [Author Response · Author response to Decision Letter 1]

25 Mar 2021

Response to reviewers 

Reviewer #1: 

The Authors have adequately replied to previous editor comments. The paper is interesting, however it's not easy to read this manuscript, the intro is too long and also data presentation is complex with a lot of data and numbers. It's unclear why the use of diuretics, the type of underlying disease (i.e CAD, VHD or DCM), the time from symptom onset to hospital admission and finally the age, didn't enter in the multivariate analysis

Dear Reviewer: Thank you for your constructive comments. We have tried to address your concerns and we found most of them as helpful to the paper improvement. 

• With regard to length of the introduction section, we have reduced some sentences which are not much related with length of hospital stay of heart failure patients. 

• With data presentation and numbers, we understand it is appropriate concern having more numerical presentation may not be attractive to the reader. But we have kept them as it is since we believe reducing the data and numerical values may not give the full picture of the extent of association of variables with patients’ length of hospital stay. 

• Concerning the variables in the multivariate analysis, we included the missed variables as per your comments. Use of diuretics is not satisfied the requirement of the multivariate analysis since it has no significant association with length of hospital stay in the univariate analysis and was not considered for multivariate analysis. Age is already included in the multivariate analysis though it was not found to have significant association. The underlying diseases included to the multivariate analysis and we did not found association with length of hospital stay by themselves. But it affected slightly the extent of association of other variables with little numerical variation without change of direction of association. Thus, the table is amended considering the new variable added to the multivariable analysis.

---

## [Decision Letter · Decision Letter 2]

6 Apr 2021

PONE-D-21-01805R2

Length of hospital stay and associated factors among heart failure patients a dmitted to the University Hospital in Northwest Ethiopia

PLOS ONE

Dear Dr. Tigabe,

Thank you for submitting your manuscript to PLOS ONE. After careful consideration, we feel that it has merit but does not fully meet PLOS ONE’s publication criteria as it currently stands. Therefore, we invite you to submit a revised version of the manuscript that addresses the points raised during the review process.

Please address to the reviewers' concerns and revise accordingly. 

We look forward to receiving your revised manuscript.

Kind regards,

Academic Editor

PLOS ONE

Reviewers' comments:

Reviewer's Responses to Questions

**Comments to the Author**

1. If the authors have adequately addressed your comments raised in a previous round of review and you feel that this manuscript is now acceptable for publication, you may indicate that here to bypass the “Comments to the Author” section, enter your conflict of interest statement in the “Confidential to Editor” section, and submit your "Accept" recommendation.

Reviewer #2: All comments have been addressed

Reviewer #3: (No Response)

2. Is the manuscript technically sound, and do the data support the conclusions?

Reviewer #2: Yes

Reviewer #3: (No Response)

3. Has the statistical analysis been performed appropriately and rigorously? 

Reviewer #2: I Don't Know

Reviewer #3: (No Response)

4. Have the authors made all data underlying the findings in their manuscript fully available?

Reviewer #2: Yes

Reviewer #3: (No Response)

5. Is the manuscript presented in an intelligible fashion and written in standard English?

Reviewer #2: Yes

Reviewer #3: (No Response)

6. Review Comments to the Author

Reviewer #2: (No Response)

Reviewer #3: We have made a review for Manuscript Number PONE-D-21-01805R2; entitled: Length of hospital stay and associated factors among heart failure patients admitted to the University Hospital in Northwest Ethiopia. The current manuscript has critical points:

1- The very limited sample size of just 263 patients and the high number of variables incorporated in the analysis compared to the small sample size. This would greatly affect the confidence with the results.

2- The authors did not incorporate important variables of heart failure assessment in modern cardiology like Echocardiography parameters, natriuretic peptides level, etc...

3- Variables described by the authors like third heart sound, etc are considered diagnostic variables of decompensated heart failure.

4- Based on the prior comments, the clinical implications of the current manuscript will not be clear and practical.

5- Extensive editing is required.

Kindest regards

7. PLOS authors have the option to publish the peer review history of their article (what does this mean?). If published, this will include your full peer review and any attached files.

Reviewer #2: **Yes: **Shady Abohashem

Reviewer #3: **Yes: **Rami Riziq Yousef Abumuaileq

---

## [Author Response · Author response to Decision Letter 2]

20 May 2021

Reviewer #3: 

We have made a review for Manuscript Number PONE-D-21-01805R2; entitled: Length of hospital stay and associated factors among heart failure patients admitted to the University Hospital in Northwest Ethiopia. The current manuscript has critical points:

1- The very limited sample size of just 263 patients and the high number of variables incorporated in the analysis compared to the small sample size. This would greatly affect the confidence with the results.

2- The authors did not incorporate important variables of heart failure assessment in modern cardiology like Echocardiography parameters, natriuretic peptides level, etc...

3- Variables described by the authors like third heart sound, etc are considered diagnostic variables of decompensated heart failure.

4- Based on the prior comments, the clinical implications of the current manuscript will not be clear and practical.

5- Extensive editing is required.

Dear Reviewer: Thank you for your constructive comments and concerns. We appreciate for your concerns.

Regarding the very limited sample size and the high number of variables incorporated in the analysis compared to the small sample size this might affect the confidence with the results. But we have tried to include all CHF patients during the study period. The prospective nature of the study design would increase the confidence of the results. Besides, we have disclosed the small size of the study may affect the generalization of the results. Thus, we have informed future researchers to consider the limitation of our study. Please note our concern was including the variables which are relevant to the length of hospital stay. However because of time and financial constraints we can’t include a large number of samples in our study. That is why these variables become large comparing to the small sample size. 

Regarding the incorporation of important variables of heart failure assessment in modern cardiology like Echocardiography parameters, natriuretic peptides level. The reason that these variables were not included in the study was such Echocardiographic parameters such as natriuretic peptides level were not done in the study setting.

Concerning the variables like third heart sound, etc are considered diagnostic variables of decompensated heart failure. In the study, not only new onset acute heart failure patients but also those who were admitted with acutely decompensated chronic heart failure patients were included, The impact of the clinical features of these patients on length of hospital stay were assessed because these variables were included in the analysis.

Regarding the clinical implications of the manuscript, we believe that the finding of study will have a role in clinical scenarios by providing data regarding the length of hospital stay and factors influencing length of hospital stay in patients with HF. Specifically, factors determine the current status of length of hospital stay and associated factors among hospitalized acute heart failure patients in the medical ward of the University of Gondar Comprehensive Specialized Hospital in Ethiopia were provided to the clinical practitioners and researchers. Moreover, clinicians would be aware of the clinical features contributing to the longer hospital stay and implementation of interventions or strategies that could reduce the heart failure patient’s hospital stay is necessary.

---

## [Decision Letter · Decision Letter 3]

27 May 2021

PONE-D-21-01805R3

Length of hospital stay and associated factors among heart failure patients a dmitted to the University Hospital in Northwest Ethiopia

PLOS ONE

Dear Dr. Tigabe,

Thank you for submitting your manuscript to PLOS ONE. After careful consideration, we feel that it has merit but does not fully meet PLOS ONE’s publication criteria as it currently stands. Therefore, we invite you to submit a revised version of the manuscript that addresses the points raised during the review process.

Please address the issues and revise accordingly.

We look forward to receiving your revised manuscript.

Kind regards,

Academic Editor

PLOS ONE

Reviewers' comments:

Reviewer's Responses to Questions

**Comments to the Author**

1. If the authors have adequately addressed your comments raised in a previous round of review and you feel that this manuscript is now acceptable for publication, you may indicate that here to bypass the “Comments to the Author” section, enter your conflict of interest statement in the “Confidential to Editor” section, and submit your "Accept" recommendation.

Reviewer #4: (No Response)

Reviewer #5: (No Response)

2. Is the manuscript technically sound, and do the data support the conclusions?

Reviewer #4: No

Reviewer #5: No

3. Has the statistical analysis been performed appropriately and rigorously? 

Reviewer #4: No

Reviewer #5: No

4. Have the authors made all data underlying the findings in their manuscript fully available?

Reviewer #4: Yes

Reviewer #5: Yes

5. Is the manuscript presented in an intelligible fashion and written in standard English?

Reviewer #4: No

Reviewer #5: No

6. Review Comments to the Author

Reviewer #4: Dr Tekle MT et al. reported clinical factors associated with length of hospital stay in patients hospitalized for acute heart failure. Unfortunately, the manuscript was poorly written, and require adequate quality of English edition. My major concern is that length of hospital stay may be influenced by different health insurance system, hospital capacity, patterns of clinical practice and physicians. Therefore, generally, this topic may lack of generalization, and need to be explored in national database or multicenter cohorts. In this regard, the interpretation of their results may be challenging. For example, their results showed that factors precipitated to acute heart failure were associated with length of hospital stay; however, various types of precipitating factors (i.e., acute coronary syndrome, infection, non-adherence etc) exist and may influence length of hospital stay. Patients with acute heart failure precipitated by acute coronary syndrome were more likely to require a longer hospital stay than those precipitated by non-adherence since these patients need a cardiac catheterization during hospital stay. Please describe your findings more precisely.

Furthermore, not surprisingly, their results showed that sicker patients with acute heart failure supported by more frequent comorbidities and/or more severe congestion were more likely to need hospital care longer. I am wondering about novel findings of the current study. If they use database from acute heart failure patients in Ethiopia as a representative of developing countries, which factor is specific for patients in developing countries.

Other concerns were raised as follows:

1/ Overall, in introduction section, interpretation of some texts was confusing. Length of hospital is a result of a hospital care. Authors described “decreased LOS results in reduced risk of medication side effects and infection, lower mortality rates, increased hospital profit with more efficient bed management, and better treatment outcomes.” However, generally, low risk of medication side effects and infection may lead to a short hospital stay.

2/ I would disagree with author description showing that a longer hospital stay was associated with poor clinical outcomes. In some countries, patients are up-titrated to maximum tolerated doses of guideline recommended heart failure medications during their hospital stay, leading to favorable outcomes. Please comment on this issue.

3/ Acute heart failure is characterized by heterogenous phenotypes, and, sometimes, it is challenging to diagnose of this cardiac syndrome. Please describe detailed information about acute heart failure definition in the methods section, citing a relevant paper.

4/ Please explain heart failure medications in detailed. The term “number of medications” is confusing. Do “medications” indicate specific treatments for heart failure or general drugs such as proton pump inhibitor and antiplatelet therapy? In addition, do “number of medications” mean number of drug tablets?

5/ Spearman is not appropriate for looking at correlation between continuous variable and categorical variable. Please remove these results or analyze it with correct statistical approach.

6/ Please draw lines between rows or columns in each table. Additionally, in Table 1, some important variables are lacking. Please describe prevalence of comorbidities or precipitating factors, which may help reader understand overall patient characteristics in these patients with acute heart failure.

7/ Again, acute coronary syndrome is, generally, a high prevalence and important precipitating factor for acute heart failure patients. Why author do not include this factor in groups of precipitating factors?

8/ How did authors select variable associated with length of hospital stay in multivariable model? With backward or forward selection?

9/Presence of elevated jugular venous pressure and/or paroxysmal nocturnal dyspnea were associated with a longer hospital stay; however, presence of III heart sound or lower admission respiratory rate were associated with a shorter hospital stay. Although all variables may express the severity of congestion, why do authors have inconsistent results?

10/ All patients were hospitalized for acute heart failure, however, there was quite low number of patients having guideline-directed heart failure medications; for example, only <25% of patients were prescribed enalapril at discharge. Surprisingly, some patients (17.5%) took metoprolol at discharge, but none took other beta-blocker drugs such as carvedilol and bisoprolol, which have been proven to be superior to metoprolol and are used in routine clinical practice. I am convinced that the different management of hospitalized heart failure may limit generalization of their findings.

11/ How about length of hospital stay across heart failure phenotypes (i.e., reduced ejection fraction, mid-range ejection fraction and preserved ejection fraction) since different ejection fraction is not only a marker of systolic function, but also has different underlying causes, management and treatments, which might lead to different length of hospital stay.

12/ Please check number of percentages in Figure 1.

13/ Please revise the manuscript adequately for an English medical journal. Overall, this paper is not relevant; for example, length of introduction section, some redundant expressions.

Reviewer #5: The authors' purpose is valuable and this should be emphasized to begin. Indeed, there are only few data available on heart failure and HF prognosis from Africa, and most are published by large, academic, universitary referal hospitals . What happens in smaller medical facilities or regional university hospitals is less known. The authors performed a large study that may be suitable for a broad readership. However there are major limitations to their study.

1 As acknowledged by the previous revievers, the manuscript is rather long, with only few,( and easy to shorten) results and conclusions. It's not easy to move from one Section to another, and to understand how relevant and useful for clinical practice are the results. Limiting the purpose to a description of the study population could be an alternate. It's interesting per se for a non African reader to know that hospital stay for acute HF may be as long as 17 days, especially in patients aged 51 years only.

2 The lack of echo and natriuretic peptides is easy to understand (BNP and NTproBNP are very expensive for low income countries). This is not a problem on my opinion.

3 The reference section may largely be improved. Data from Poland, Japan, China, Germany are not required, I believe. By contrast, the authors omitted a major study on HF from Subsaharan Africa , the Thesus-Hf study ( The sub-Saharan Africa Survey of Heart Failure (THESUS-HF) prospective cohort study).

4 I believe we should sustain and promote African studies especially from centers who make efforts to perform academic research. May be a way to make this submission suitable for publication would be to largely shorten it and to limit results to a description of the real world description of HF in their institution.

7. PLOS authors have the option to publish the peer review history of their article (what does this mean?). If published, this will include your full peer review and any attached files.

Reviewer #4: **Yes: **Masatake Kobayashi

Reviewer #5: **Yes: **Jean-Jacques Monsuez

---

## [Author Response · Author response to Decision Letter 3]

1 Jan 2022

Dear Reviewers thank you for your constructive comments. We have tried to address your concerns and we found most of them as helpful to the paper improvement.

---

## [Decision Letter · Decision Letter 4]

17 Jan 2022

PONE-D-21-01805R4Length of hospital stay and associated factors among heart failure patients a dmitted to the University Hospital in Northwest EthiopiaPLOS ONE

Dear Dr. Tigabe,

Thank you for submitting your manuscript to PLOS ONE. After careful consideration, we feel that it has merit but does not fully meet PLOS ONE’s publication criteria as it currently stands. Therefore, we invite you to submit a revised version of the manuscript that addresses the points raised during the review process.

Please revise and address the issues.

We look forward to receiving your revised manuscript.

Kind regards,

Academic Editor

PLOS ONE

Reviewers' comments:

Reviewer's Responses to Questions

**Comments to the Author**

1. If the authors have adequately addressed your comments raised in a previous round of review and you feel that this manuscript is now acceptable for publication, you may indicate that here to bypass the “Comments to the Author” section, enter your conflict of interest statement in the “Confidential to Editor” section, and submit your "Accept" recommendation.

Reviewer #3: (No Response)

Reviewer #4: (No Response)

2. Is the manuscript technically sound, and do the data support the conclusions?

Reviewer #3: (No Response)

Reviewer #4: No

3. Has the statistical analysis been performed appropriately and rigorously? 

Reviewer #3: (No Response)

Reviewer #4: Yes

4. Have the authors made all data underlying the findings in their manuscript fully available?

Reviewer #3: (No Response)

Reviewer #4: Yes

5. Is the manuscript presented in an intelligible fashion and written in standard English?

Reviewer #3: (No Response)

Reviewer #4: No

6. Review Comments to the Author

Reviewer #3: (No Response)

Reviewer #4: Unfortunately, my concerns were not addressed, and I cannot see what point authors modify or correct exactly in their revised version of manuscript. As authors can see in my comments, a variable, “precipitating factors”, was so confusing since clinical meaning and/or prognostic value varied greatly among precipitating factors including, acute coronary syndrome, infection, arrhythmia, and poor adherence. Even though this variable was statistically associated with length of hospital stay, I cannot interpret this result.

Sadly, although authors tried to argue my concerns, they did not modify their manuscript and explain them in the “response to reviewers”. I am convinced that readers also have similar concerns, and these raised points should be addressed adequately, particularly, in the limitations section.

Additionally, I am not sure whether the revised manuscript was edited by medical English editing. For example, their description in the results section entitled, “predictors of length of hospital stay” included many redundancies.

Furthermore, in this study, heart failure was defined per the ESC guidelines; however, in their description table, authors did not show any findings of electrocardiogram, chest x-ray, and/or echocardiogram, or values of natriuretic peptide. Please present these variables.

7. PLOS authors have the option to publish the peer review history of their article (what does this mean?). If published, this will include your full peer review and any attached files.

Reviewer #3: **Yes: **Rami Riziq Yousef Abumuaileq

Reviewer #4: No

---

## [Author Response · Author response to Decision Letter 4]

23 Mar 2022

Dear Reviewer # 4 Thank you for your constructive comments and concerns. We have tried to address your concerns and we found most of them as helpful to the paper improvement. First of all, for the crucial comments that you have given to our paper, we would like to provide our deepest heart felt gratitude.

---

## [Decision Letter · Decision Letter 5]

29 Mar 2022

PONE-D-21-01805R5Length of hospital stay and associated factors among heart failure patients a dmitted to the University Hospital in Northwest EthiopiaPLOS ONE

Dear Dr. Tigabe,

Thank you for submitting your manuscript to PLOS ONE. After careful consideration, we feel that it has merit but does not fully meet PLOS ONE’s publication criteria as it currently stands. Therefore, we invite you to submit a revised version of the manuscript that addresses the points raised during the review process.

Please revise. 

We look forward to receiving your revised manuscript.

Kind regards,

Academic Editor

PLOS ONE

Reviewers' comments:

Reviewer's Responses to Questions

**Comments to the Author**

1. If the authors have adequately addressed your comments raised in a previous round of review and you feel that this manuscript is now acceptable for publication, you may indicate that here to bypass the “Comments to the Author” section, enter your conflict of interest statement in the “Confidential to Editor” section, and submit your "Accept" recommendation.

Reviewer #3: (No Response)

Reviewer #4: All comments have been addressed

Reviewer #6: All comments have been addressed

2. Is the manuscript technically sound, and do the data support the conclusions?

Reviewer #3: (No Response)

Reviewer #4: Yes

Reviewer #6: Partly

3. Has the statistical analysis been performed appropriately and rigorously? 

Reviewer #3: (No Response)

Reviewer #4: Yes

Reviewer #6: No

4. Have the authors made all data underlying the findings in their manuscript fully available?

Reviewer #3: (No Response)

Reviewer #4: Yes

Reviewer #6: Yes

5. Is the manuscript presented in an intelligible fashion and written in standard English?

Reviewer #3: (No Response)

Reviewer #4: Yes

Reviewer #6: Yes

6. Review Comments to the Author

Reviewer #3: (No Response)

Reviewer #4: (No Response)

Reviewer #6: A significant problem with the current version of the paper is the statistical analysis. First, the use of multivariate analysis is not properly mentioned in the Methods. Second, most of the variables in linear regression (see Table 3) are qualitative, which significantly affects its efficiency. It would be more logical to use other types of regression analysis, taking into account the type of input data. In any case, it is necessary to explain the reasons for choosing the multiple analysis method in Methods by discussing its limitations in the context of the peculiarities of the data. Moreover, very many indicators are included in the analysis with a fairly limited number of observations. I am not sure if presented results are correct. The principles of selection of indicators in the multivariate model, the modeling procedure need to be clarified and, most likely, corrected.

It is unclear why the following phrases are given for a number of variables in Table 3: «causes of AHF» or « precipitating factor of AHF». Perhaps this should be reflected in the Methods when describing the data concept, or in the Discussion.

7. PLOS authors have the option to publish the peer review history of their article (what does this mean?). If published, this will include your full peer review and any attached files.

Reviewer #3: No

Reviewer #4: No

Reviewer #6: No

---

## [Author Response · Author response to Decision Letter 5]

23 May 2022

Dear Reviewer # 6: thank you for your constructive comments and concerns which are helpful for our paper quality. 

based on your comments we have made modifications to our manuscript.

---

## [Decision Letter · Decision Letter 6]

20 Jun 2022

Length of hospital stay and associated factors among heart failure patients a dmitted to the University Hospital in Northwest Ethiopia

PONE-D-21-01805R6

Dear Dr. Tigabe,

We’re pleased to inform you that your manuscript has been judged scientifically suitable for publication and will be formally accepted for publication once it meets all outstanding technical requirements.

Kind regards,

Academic Editor

PLOS ONE

Additional Editor Comments (optional):

Reviewers' comments:

Reviewer's Responses to Questions

**Comments to the Author**

1. If the authors have adequately addressed your comments raised in a previous round of review and you feel that this manuscript is now acceptable for publication, you may indicate that here to bypass the “Comments to the Author” section, enter your conflict of interest statement in the “Confidential to Editor” section, and submit your "Accept" recommendation.

Reviewer #3: (No Response)

Reviewer #6: All comments have been addressed

2. Is the manuscript technically sound, and do the data support the conclusions?

Reviewer #3: No

Reviewer #6: Yes

3. Has the statistical analysis been performed appropriately and rigorously? 

Reviewer #3: No

Reviewer #6: Yes

4. Have the authors made all data underlying the findings in their manuscript fully available?

Reviewer #3: Yes

Reviewer #6: Yes

5. Is the manuscript presented in an intelligible fashion and written in standard English?

Reviewer #3: No

Reviewer #6: Yes

6. Review Comments to the Author

Reviewer #3: The authors have superficially addressed the previous comments, however the main defects are still present. The limited sample size, doubts regarding the diagnosis, management and proper follow up in a hospital without a specialized cardiology department.

Kind regards

Reviewer #6: Although I believe it is appropriate to perform multivariate data analysis using somewhat different methods, I agree with the authors' position, although their approach has a number of limitations.

7. PLOS authors have the option to publish the peer review history of their article (what does this mean?). If published, this will include your full peer review and any attached files.

Reviewer #3: **Yes: **Rami Riziq Yousef Abumuaileq

Reviewer #6: No

---

## [Editor Report · Acceptance letter]

13 Jul 2022

PONE-D-21-01805R6 

*Length of hospital stay and associated factors among heart failure patients admitted to the University Hospital in Northwest Ethiopia*

Dear Dr. Tigabe Tekle:

I'm pleased to inform you that your manuscript has been deemed suitable for publication in PLOS ONE. Congratulations! Your manuscript is now with our production department. 

Kind regards, 

on behalf of

Dr. Robert Jeenchen Chen 

Academic Editor

PLOS ONE